# Targeting *Plasmodium* Life Cycle with Novel Parasite Ligands as Vaccine Antigens

**DOI:** 10.3390/vaccines12050484

**Published:** 2024-04-30

**Authors:** Shan Khan, Manas Paresh Patel, Aleem Damji Patni, Sung-Jae Cha

**Affiliations:** Department of Medical Sciences, Mercer University School of Medicine, 1501 Mercer University Drive, Macon, GA 31207, USA; shan.khan@live.mercer.edu (S.K.); manas.paresh.patel@live.mercer.edu (M.P.P.); aleem.damji.patni@live.mercer.edu (A.D.P.)

**Keywords:** malaria, ligand-receptor interaction, phage display, vaccine target

## Abstract

The WHO reported an estimated 249 million malaria cases and 608,000 malaria deaths in 85 countries in 2022. A total of 94% of malaria deaths occurred in Africa, 80% of which were children under 5. In other words, one child dies every minute from malaria. The RTS,S/AS01 malaria vaccine, which uses the *Plasmodium falciparum* circumsporozoite protein (CSP) to target sporozoite infection of the liver, achieved modest efficacy. The Malaria Vaccine Implementation Program (MVIP), coordinated by the WHO and completed at the end of 2023, found that immunization reduced mortality by only 13%. To further reduce malaria death, the development of a more effective malaria vaccine is a high priority. Three malaria vaccine targets being considered are the sporozoite liver infection (pre-erythrocytic stage), the merozoite red blood cell infection (asexual erythrocytic stage), and the gamete/zygote mosquito infection (sexual/transmission stage). These targets involve specific ligand-receptor interactions. However, most current malaria vaccine candidates that target two major parasite population bottlenecks, liver infection, and mosquito midgut infection, do not focus on such parasite ligands. Here, we evaluate the potential of newly identified parasite ligands with a phage peptide-display technique as novel malaria vaccine antigens.

## 1. Introduction

Malaria is the deadliest known vector-borne infectious disease. Malaria can be controlled by killing mosquitoes with insecticides, drug treatment of patients, and vaccination of humans. Although insecticides and drugs have reduced malaria death, this approach faces difficulties due to a selection of resistant mosquitoes and parasites. Moreover, an effective malaria vaccine is yet to be implemented. The malaria parasite has a complex life cycle with multiple stages in the mosquito and in humans, with each step involving specific host cell recognition. Therefore, targeting the [parasite ligand-host cell receptor] pair is a prime goal of malaria vaccine development.

Human infection starts with the release of ~50 sporozoites into the human skin by the bite of an infected *Anopheles* mosquito [1,2]. Sporozoites enter the blood circulation and move to the first target organ, the liver. Circulating sporozoites are retained by the liver via electrostatic interaction between the positively charged CSP protein that covers the sporozoite surface and the negatively charged glycosaminoglycans (GAGs) produced by Stellate cells in the liver parenchyma and protruding through the fenestrations of the liver sinusoidal (blood vessel) lining [3]. Next, sporozoites traverse the liver sinusoidal lining by invading macrophage-like Kupffer cells or endothelial cells [4,5], followed by an invasion of a hepatocyte to establish a productive infection. Although only a few sporozoites reach their destination, each infected liver cell generates around 30,000 hepatic merozoites which are released into the blood circulation where they proliferate and cause pathogenesis [2,6]. Sporozoite liver infection constitutes the most dramatic malaria parasite population bottleneck and has been a focus of malaria vaccine development. The most advanced malaria vaccine RTS,S/AS01 targets the sporozoite CSP, which is essential for liver infection [7,8]. Malaria vaccines that target liver infection (before RBC infection) are called pre-erythrocytic vaccines (PEV), and vaccines that target asexual blood-stage parasite antigens, mainly on the merozoites surface, are called blood-stage vaccines (BSV). Since the blood stage starts with the release of a large number of hepatic merozoites, followed by quick (30~70 s) invasion of RBCs utilizing redundant ligand-receptor pairs [9], the development of blood-stage vaccines is more challenging. Indeed, no sterile protection has been reported from human BSV trials [7]. In addition to liver infection, another parasite population bottleneck occurs at the initial stages of mosquito infection [10,11]. In an infected human, less than 5% of *Plasmodium* blood-stage parasites are committed to developing into sexual-stage gametocytes [12], the only form that can survive in the mosquito. In the blood bolus, male and female gametocytes are rapidly activated into gametes. One male gametocyte generates eight microgametes after three rounds of DNA replication. Microgametes actively move in search of activated female macrogametes for fertilization and the formation of zygotes. Zygotes differentiate into motile ookinetes that move in the blood bolus and invade the mosquito midgut to form oocysts, lodged between the midgut epithelium and the basal lamina. Each oocyst generates thousands of sporozoites that, when released, invade the mosquito salivary glands from where they are transmitted to a new human host. Out of thousands of ingested gametocytes, only a few oocysts form [11], defining a strong bottleneck. Malaria vaccines that target mosquito-stage parasite development are called transmission-blocking vaccines (TBV).

Transition through bottlenecks involves specific cell-cell recognition steps. For liver infection, sporozoites specifically interact with the phagocytic Kupffer cells that line the sinusoids to exit the circulation and then specifically infect hepatocytes and no other cell type. In the mosquito, fertilization requires specific recognition between male and female gametes and ookinete traversal involves its recognition of mosquito midgut epithelium receptors. Using a phage-peptide display approach, [parasite ligand-host cell receptor] pairs involved in these parasite bottlenecks have been identified [13,14,15,16]. Interestingly, many of the parasite surface ligands are conserved moonlighting proteins that have acquired separate ligand functions. This review provides an overview of these *P. falciparum* parasite ligands and evaluates their potential for the development of novel malaria vaccine antigens.

## 2. Pre-Erythrocytic *Plasmodium* Ligands

*Plasmodium* sporozoites interact with Kupffer cells to leave the circulation and enter the liver parenchyma [4]. Sporozoite surface glyceraldehyde 3-phosphate dehydrogenase (GAPDH) was identified as a ligand for Kupffer cell traversal [15]. In the liver parenchyma, sporozoites specifically infect hepatocytes using the phospholipid scramblase (PLSCR) protein as a ligand [14].

### 2.1. Sporozoite Exit from the Circulation

A phage peptide-display library that displays 12 amino acid random peptides fused to the coat protein pVIII of the M13 filamentous phage was screened against a primary rat Kupffer cell culture [17,18]. The selected Kupffer cell-binding P39 peptide (DCAIVYAYDPCL) competitively inhibited sporozoite-Kupffer cell interaction in vitro. This suggested that the P39 peptide binds to a Kupffer cell receptor for sporozoite interaction. Far-Western blotting using the P39 peptide as a probe and mass spectrometry assays identified the Kupffer cell CD68 protein as the P39 peptide binding target. Confirming this finding, sporozoite liver infection is significantly reduced in CD68KO mice [18]. These findings suggested that P39 is a structural mimic of a sporozoite ligand for CD68 interaction. To test this hypothesis, antibodies against the P39 peptide were generated. These antibodies were bound to a ~40 kDa sporozoite surface molecule [15]. Immunoprecipitation and mass spectrometry assays identified the ~40 kDa sporozoite surface protein *Plasmodium* GAPDH as a ligand for CD68 interaction. Follow-up studies found that anti-P39 antibodies bind to two epitopes in the GAPDH C-terminal end and antibodies against these epitopes inhibited sporozoite liver infection in vivo [19]. 

The canonical role of GAPDH is to facilitate the conversion of glyceraldehyde-3-phosphate into 1–3 di-phosphoglycerate during glycolysis [20]. Although GAPDH acts as a quintessential housekeeping protein for cell energy production across many species, research identified GAPDH as a moonlighting protein whose function extends well beyond its traditional role in glycolysis. GAPDH is involved in DNA repair, gene expression control, cell signaling, and interaction with other molecules such as RNA and protein. Furthermore, diverse infectious microbes expose GAPDH onto their surface and utilize it for attachment to host cell protein, extracellular matrix degradation, modification of cell signaling, immune evasion, and virulence factor [20,21,22]. These properties have spotlighted GAPDH as a vaccine target for bacterial and parasitic diseases. For instance, immunization with recombinant GAPDH of *Erysipelothrix rhusiopathiae*, the causative agent of animal erysipelas and human erysipeloid, elicited sterile protection in a mouse model [23]. Immunization with recombinant GAPDH of *Streptococcus dysgalactiae*, a bacterial pathogen causing intramammary infections in dairy bovines, improved the survival rates and decreased the bacterial burdens in the mammary glands in immunized mice [24]. Conserved GAPDH peptides among multiple pathogens have been explored for a universal vaccine antigen that targets *Listeria monocytogenes*, *Mycobacterium marinum*, and *Streptococcus pneumoniae* [25,26]. On the other hand, predicted GAPDH B-cell epitopes and pyruvate dehydrogenase E1 subunit alpha in *Streptococcus iniae*, a bacterial flounder pathogen, were fused for a multiple-epitope vaccine antigen, which increased survival rate in immunized animals compared to single-antigen immunized animals [27]. Multiple-epitope vaccines using *Schistosoma mansoni* GAPDH and Sm10-DLC (a 10-kDa dynein light chain protein) peptides showed protection in a mouse model [28]. The challenge, however, lies in distinguishing between the host and pathogen GAPDH proteins to avoid potential autoimmune reactions, necessitating the identification of protective epitopes unique to the pathogen protein. 

Four major requirements of potential vaccine antigens are: (1) to contain the sequence located within the ligand region that interacts with the target receptor, (2) to have low genetic polymorphism, (3) not to be too similar to the host homologs, and (4) to have immunogenicity that generates protective efficacy. *GAPDH* genes among human-infecting *Plasmodium* species are well-conserved at the level of 87% identity [15]. However, *Plasmodium* GAPDH identity, with its human counterpart, is lower, at the level of 64% identity (Table 1). Genetic polymorphism is low, with only one non-synonymous single nucleotide polymorphism (NS-SNP) having been identified out of 275 *P. falciparum* strains (Table 1). There are nine parasite-specific GAPDH peptides that are well-conserved among *Plasmodium* species and divergent from mammalian orthologs [15]. Interestingly, two *Plasmodium*-specific GAPDH peptides were identified as epitopes for Kupffer cell interaction [19]. Antibodies against these epitope peptides do not cross-react with the mammalian GAPDH. Indeed, these antibodies significantly inhibited sporozoite liver infection in mice. One of these peptides, G6-1-20, overlaps with a parasite-specific B-cell epitope (GAPDH 283–292, see Table 2). Therefore, parasite-specific GAPDH epitope peptides are candidates for novel malaria vaccine antigens. Since these epitopes are small (20 amino acids), they could be fused to existing malaria vaccine antigens for additive or synergistic protective efficacy [29]. 

### 2.2. Hepatocyte Infection

The same phage-peptide display library was used to screen for peptides that bind to freshly isolated mouse hepatocytes [14]. Out of five candidates, the HP1 peptide (NCTSNDIWESCT) was the strongest hepatocyte-binding peptide, and this peptide competitively inhibited sporozoite-hepatocyte interaction in vitro [14]. The basic hypothesis was that HP1 is a structural mimic of a sporozoite ligand for hepatocyte interaction and competes with the sporozoite for binding to the hepatocyte receptor. Supporting this hypothesis, anti-HP1 antibodies recognized molecules on the surface of unpermeabilized *P. berghei* and *P. falciparum* sporozoites. On Western blots of *P. berghei* sporozoite lysates, anti-HP1 antibodies bound to a ~50 kDa sporozoite protein and mass spectrometry identified 6 candidate sporozoite proteins of this size range. All 6 candidates were expressed as recombinant proteins, of which anti-HP1 antibodies recognized only PbPLSCR. In parallel, a hepatocyte membrane protein for HP1 peptide binding was identified as Carbamoyl-phosphate synthase 1 (Cps1). Interaction of PbPLSCR with Cps1 was confirmed with pull-down and ELISA assays, leading to the identification of *Plasmodium* PLSCR and hepatocyte Cps1 as sporozoite ligand and hepatocyte receptor proteins, respectively.

PLSCR is a type II membrane protein, whose N’-terminal region is predicted to face the cytosolic space, with the C-terminal end to be embedded in the membrane. PLSCR catalyzes the scrambling of asymmetrically polarized plasma membrane phospholipid components in a Ca^2+^-dependent and ATP-independent manner [30,31]. Mammalian cell membranes are enriched for sphingomyelin and phosphatidylcholine in the outer leaflet and enriched for phosphatidylserine (PS), phosphatidylethanolamine, and phosphatidylinositol in the inner leaflet. PLSCRs activate the externalization of PS, which activates blood coagulation and apoptosis. There are three phospholipid scrambling enzyme families: the PLSCR family, the TMEM16 family, and the XKR family [30]. Only the PLSCR family has structural similarity with Tubby-like proteins which possess a 12-stranded beta barrel surrounding a central alpha helix. *Plasmodium* PLSCR has the conserved structure of the PLSCR family [32] and its scramblase activity was confirmed; however, PfPLSCR is dispensable for blood-stage growth [33]. Apart from these canonical activities, PLSCR has additional functions as a moonlighting protein [34]. PLSCR can be secreted into the extracellular matrix (ECM) or enter the nucleus. In the cytosolic space, PLSCRs interact with other proteins involved in cell death, autophagy, inflammation, molecular trafficking, and gene regulation. In the extracellular space or endosomal compartment, PLSCRs interact with viruses or other cellular receptor molecules for internalization. Some enveloped viruses utilize canonical scramblase function to induce PS exposure on the host cell membrane, which serves as the viral envelope that induces apoptotic mimicry that facilitates viral entry through host cell phagocytosis. On the other hand, human PLSCR1 targets SARS-CoV-2-containing vesicles and inhibits virus entry and membrane fusion by the spike protein through its ectodomain (β-barrels and C-terminal helix), which has no direct correlation with scramblase activity [35].

PfPLSCR has the potential for a PEV antigen since it is highly expressed on the sporozoite surface [14,29,33,36,37]. As the human PLSCR1 ectodomain (which is exposed at the extracellular surface) acts as a moonlighting protein for inhibition of SARS-CoV-2 entry, the PfPLSCR ectodomain (which is exposed at the parasite surface and has no correlation with scramblase activity) is expected to act as a ligand for hepatocyte interaction. Therefore, only one predicted *Plasmodium*-specific B-cell epitope (MGFKLDFN, Table 2) in the ectodomain is expected to generate protective antibodies. The other epitope peptides are expected to be located in the cytosolic compartment. It is expected that the ligand epitope in the ectodomain may contribute for protective immunity when used as an antigen. Except for the *Plasmodium* sporozoite, PLSCR has never been investigated as a ligand of infectious microbes or a vaccine candidate. *Plasmodium* has one PLSCR gene, and PLSCR genes among *Plasmodium* species are well-conserved (~75% identity) [14]; however, the human genome has 5 PLSCR genes and 10 isoforms (https://uniprot.org (accessed on 28 March 2024)), of which maximum identity reaches 21% when compared with human PLSCR3. Moreover, only two non-synonymous PfPLSCR SNPs have been reported out of 301 sequenced strains (Table 1). Divergent sequence identity from mammalian host proteins and low polymorphism in the field favor the potential use of PfPLSCR as an additional malaria vaccine antigen. Indeed, antibodies against PbPLSCR inhibited both *P. berghei* in vivo and *P. falciparum* sporozoite-hepatocyte interaction in vitro [14]. Genetic polymorphism of a vaccine antigen can dampen enthusiasm for its general application. PfCSP, the most prominent malaria vaccine antigen, has 48 NS-SNPs out of 310 sequenced strains and 14% of mutant alleles have frequencies over 30%. On the other hand, two SNPs in PfPLSCR were identified in the cytosolic N’-terminal region with 1% frequency (https://plasmodb.org (accessed on 28 March 2024)).

## 3. *Plasmodium* Ligands in the Mosquito

The first specific parasite interaction in the mosquito midgut occurs between male and female gametes at fertilization. Female gamete heat-shock protein 90 (HSP90) was identified as a ligand for fertilization [16]. The zygote then develops into an ookinete that moves to the mosquito midgut epithelium. Ookinetes traverse the mosquito midgut using surface enolase as a ligand [13] and then differentiate into oocysts that produce thousands of sporozoites. 

### 3.1. Fertilization

Purified male *P. berghei* gametocytes were incubated with a phage-peptide display library for selecting male gamete-binding peptides. The MG1 peptide (SFRSRLQPYSCA) was selected and it competitively inhibited *P. berghei* and *P. falciparum* fertilization, both in vitro and in the mosquito midgut [16]. The hypothesis that MG1 peptide is a structural mimic of a female gamete protein for fertilization was confirmed with assays using anti-MG1 antibodies. Anti-MG1 antibodies bound to the female gamete surface and recognized a ~110 kDa female gamete membrane protein. Mass spectrometry assay identified three candidate proteins in this size range that were expressed as recombinant proteins. Anti-MG1 antibodies recognized *Plasmodium* heat-shock protein 90 (HSP90) only. Immuno-fluorescence assays confirmed surface exposure of HSP90 on *P. berghei* and *P. falciparum* female gametes. Moreover, anti-HSP90 antibodies inhibited gamete fertilization in the mosquito midgut confirming that *Plasmodium* female gamete HSP90 as a fertilization ligand. 

HSP90 is a well-known molecular chaperone that plays crucial roles in various cellular processes, including signal transduction, protein folding, and response to stress conditions. Canonical roles of HSP90 include facilitating protein folding, stabilization, and activation of a wide range of client proteins [38]. HSP90 operates as a homodimer whose conformation is regulated by ATP binding to its N’-terminal ATPase domain. The binding of various co-chaperones can modulate its activity. HSP90 function is not limited to the intracellular environment; extracellular HSP90 also plays a role in immune responses by interacting with surface receptors on immune cells, thereby modulating cellular functions such as cytokine secretion, cell maturation, and antigen presentation. HSP90 has been explored as an adjuvant since it activates and modulates professional antigen-presenting cells to induce robust immune responses [39]. As is the case for *Plasmodium*, mammalian oocyte surface HSP90s are important for fertilization as well [40,41]. Surface HSP90 exposure on diverse infectious microorganisms has been reported, which suggests HSP90 may be a vaccine candidate. Immunoblotting assays using convalescent sera from *Streptococcus oralis*-infected patients detected two immunodominant antigens. An 85 kDa protein was identified as HSP90 and antibodies against HSP90 protected immunized animals [42]. HSP90 of the oriental liver fluke, *Clonorchis sinensis*, was assessed for T-cell activation and antibody production [43]. Antisera against human fungal disease *Aspergillus fumigatus* identified enolase and HSP90 as vaccine candidates [44]. Proteome analysis using antisera against *Borrelia burgdorferi*, a causative agent of Lyme disease, identified HSP90 as an immunodominant antigen [45]. Two HSP90 epitopes of *Candida albicans*, an opportunistic pathogenic yeast, have been identified and applied as vaccine antigens for protective immunity [46,47].

*P. falciparum* has one HSP90 gene and humans have two genes and three isoforms, of which the maximum *Plasmodium*-human identity is 75% (HsHSP90B, Table 1). Seven non-synonymous SNPs were identified out of 301 sequenced *P. falciparum* strains, of which 6 NS-SNPs were found in the predicted B-cell epitope PfHSP90_211–340 (Table 2). *Plasmodium* HSP90 is recognized as a potential vaccine candidate since it is exposed on the parasite surface during diverse stages, even on the infected RBC surface, which is important for fertilization and essential for blood-stage parasite growth [40,41,48,49,50,51,52]. Antibodies against *P. berghei* HSP90 inhibited parasite fertilization, which shows the potential of *Plasmodium* HSP90 as a TBV antigen [16]. Anti-sporozoite and anti-asexual blood stage parasite activity of these antibodies is yet to be assessed. Since HSP90 acts as a chaperon molecule in diverse pathways and surface exposure of HSP90 was confirmed at multiple parasite stages [48,49,50], the utility of anti-HSP90 antibodies may extend beyond the TBV. Despite PfHSP90’s high sequence identity with human orthologs, there are over 20 divergent peptides with an identity lower than 50%, of which four of them are predicted to be B-cell epitopes (Table 2). 

### 3.2. Traversal of the Midgut Epithelium

Screening for mosquito midgut-binding peptides using a phage-peptide display library led to the identification of the *Plasmodium* ookinete surface enolase as a ligand for mosquito midgut invasion [53]. The selected SM1 peptide (PCQRAIFQSICN) bound to the luminal side of the mosquito midgut and prevented interaction with ookinetes. This suggested that the SM1 peptide binds to a receptor on the mosquito midgut for ookinete invasion and that the SM1 peptide mimics the conformation of an ookinete ligand for midgut interaction. This hypothesis was confirmed with a series of assays using anti-SM1 antibodies. Immunofluorescence assays showed that anti-SM1 antibodies bind to both *P. berghei* and *P. falciparum* ookinete surfaces. A Western blotting assay found that the anti-SM1 antibodies bound to two, ~65 kDa and ~48 kDa, major protein bands. Mass spectrometry assay identified the 65 kDa protein as RNA helicase, which is a cytoplasmic protein, and the 48 kDa protein as enolase. Recombinant *P. falciparum* enolase was generated for the production of antibodies in rabbits. Immuno-electron microscopy confirmed the surface localization of *P. berghei* ookinete enolase. Importantly, anti-enolase antibodies inhibited *P. berghei* and *P. falciparum* ookinete midgut invasion [13]. 

The main function of enolase is the catalysis of 2-phosphoglycerate to phosphoenolpyruvate, reordering the single bond between carbon atoms to form a higher energy double bond in glycolysis [54]. However, as a moonlighting protein, enolase plays an important role in diverse pathogenic microorganisms for adhering to the ECM of various organisms [54,55]. As an adherence factor, enolase interacts with host ECM and recruits mammalian plasminogen, which in turn plays a key role in the fibrinolytic system maintaining homeostasis via the creation and dissolution of fibrin clots. Plasminogen should be activated into plasmin for fibrinolysis and plasminogen binding to enolase promotes this activation, thus allowing pathogens to traverse the host ECM [56]. Plasminogen-enolase interaction plays an important role in the *Plasmodium* ookinete mosquito midgut invasion. In addition to serving as a mosquito midgut ligand, ookinete surface enolase mediates the binding of blood plasminogen via a C-terminal 6-amino acid motif (DKSLVK) that is recognized by a lysine-binding Kringle domain of the plasminogen. Confirming this role, feeding an infectious blood meal with excess soluble 6-aa peptide competitively inhibited ookinete mosquito midgut invasion by preventing plasminogen binding to ookinete surface enolase [13]. 

Enolase has been investigated as a vaccine candidate to protect from diverse infectious microbes. The avian bacterial pathogen, *Mycoplasma synoviae*, uses enolase as an adherence factor. Immunization and challenge assayusing six major immunogenic antigens identified that enolase elicited the best protective efficacy in challenged chickens [57]. This trend was also seen in ticks, where it was shown that anti-enolase antibodies reduced tick attachment [58]. The bovine disease-causing bacterium *Anaplasmasa marginale* uses enolase for attachment to the host erythrocyte surface. *A. marginale* AmEno01 protein was identified as a vaccine candidate, as it exists across all groups of *A. marginale* strains with well-conserved 3D structures [59]. Bioinformatics studies of enolase in *Trypanosoma cruzi*, a causative agent of American trypanosomiasis, identified a consensus sequence of the enolase protein from 15 *T. cruzi* strains as a vaccine candidate [60]. 

*Plasmodium* enolase is a promising TBV antigen candidate since it is exposed on the ookinete surface for interaction with a mosquito midgut receptor and is important for the movement of ookinete in the blood bolus using the fibrinolysis pathway. Indeed, anti-enolase antibodies inhibited ookinete mosquito invasion. Interestingly, treatment of 6-aa peptide for a lysine-binding Kringle domain competitively inhibited parasite mosquito midgut invasion with similar efficacy as antibody treatment, which suggests that antibody against the 6-aa peptide can inhibit ookinete midgut invasion [13]. Due to its small size, the 6-aa peptide can be fused with an existing antigen for immunization. 

## 4. Conclusions and Perspectives

All merozoite ligands for RBC infection identified so far are parasite-specific proteins [9]. In contrast, all parasite ligands at two major population bottlenecks are moonlighting proteins acting uncoupled from their canonical roles. In terms of vaccine development, concerns are not related to their multifunctional properties, but to their conserved protein structures and sequences with host protein orthologues. One hypothesis for why infectious microorganisms use conserved moonlighting proteins as ligands is that the host will not elicit antibodies against these conserved proteins. Indeed, most of the predicted linear B-cell epitopes in the microbes and human protein orthologues overlap [61]. However, the parasite ligand epitope peptides for host cell recognition may not be located only in highly conserved regions. Two *Plasmodium* sporozoite GAPDH epitopes, G6-1-20 and G6-41-60, which are peptides for CD68 interaction, have 70% and 65% identities with host proteins and antibodies against these epitope peptides do not cross-react with the host protein [19]. Although these epitope peptides are not located in the immunodominant regions, immunization of each epitope peptide elicited enough antibodies for protective immunity. Therefore, epitope-based vaccination strategies will resolve conserved antigen issues and will enable easy fusion of multiple epitope antigens for increased protective efficacy. 

A major weakness of TBVs is that transmission-blocking antibodies do not protect the immunized hosts. Thus, the addition to the TBV of an antigen that confers protective immunity to the vaccinated person is highly desirable. Optimizing an immunization regimen for two different vaccine antigens has many hurdles. The molecular size of the most advanced PEV (RTS,S/AS01E) or the most advanced TBV (P230) is large enough to generate broad-range polyclonal antibodies that can ‘dilute’ protection efficiency by generating antibodies unrelated to protection [7,62]. Immunization with two large antigens may elicit competition and immunodominance of antibodies irrelevant to protection. The development of fusion vaccine antigens containing multiple epitopes targeting both liver infection and mosquito transmission will improve malaria vaccine strategies to prevent both infection of humans and transmission to mosquitoes. Of note, parasite ligands in the two major population bottlenecks were identified through host cell binding peptide selection. All the selected peptides were identified as structural mimics of parasite ligands. Immunization with these peptides elicited protective antibodies, which strongly implies the feasibility of using these peptides, instead of parasite-host antigens, as vaccine antigens. 

## Figures and Tables

**Table 1 vaccines-12-00484-t001:** Major genetic characteristics of *P. falciparum* ligand genes as malaria vaccine antigens. Genetic variations of parasite proteins were identified with NS-SNPs. Each NS-SNP is denoted for location, amino acid change, and percent of the minor allele among all sequenced strains (https://plasmodb.org (accessed on 28 March 2024)). The maximum % identity between parasites and all human isoforms was determined.

Ligand	*P. falciparum* Gene				Human Gene	
	# Gene, Isoforms	Amino Acids	Genetic Variation		# Gene, Isoforms	Maximum Identity (%)
			NS-SNPs (%)	Strains		
GAPDH	1, 1	337	140Q-K (33)	275	1, 2	64
PLSCR	1, 1	275	50S-N (1>), 52M-I (1)	301	5, 10	21
HSP90	1, 1	745	58A-S (2), 229G-R (4), 231R-E (3), 233G-E (1), 235E-G (6), 239K-E (25), 256N-K (1)	301	2, 3	75
Enolase	1, 1	446	301V-I (1)	300	4, 10	69

**Table 2 vaccines-12-00484-t002:** In silico prediction of linear B-cell epitope peptides for searching parasite-specific peptides in each *P. falciparum* ligand protein. Percent identity denotes the maximum identity between parasite and human protein orthologs. Parasite epitope peptides no shorter than eight amino acids and are listed on no higher identity than 60%. B-cell epitopes were predicted using http://tools.iedb.org/bcell/ (accessed on 28 March 2024).

Ligand	Position	Peptide Sequence	% Identity
GAPDH	102–111	FLTKELASSH	0
	189–199	VDGPSKGGKDW	0
	283–292	EVVSQDFVHD	60
PLSCR	5–26	NIHMQPNINYSYRNPNMYNMNY	0
	35–43	QQQMQLFVN	0
	72–79	MGFKLDFN	0
HSP90	156–169	FTVTKDETNEKLGR	0
	211–340	RQNEKEITASEEEEGEGEGEREGEEEEEKKKKTGEDKNADESKEENEDEEKKEDNEEDDNKTDHPKVEDVTEELENAEKKKKEKRKKKIHTVEHEWEELNKQKPLWMRKPEEVTNEEYASFYKSLTNDWE	28
	562–590	CCTKEGLDIDDSEEAKKDFETLKAEYEGL	48
	716–742	SIDEEENNDIDLPPLEETVDATDSKME	33
Enolase	83–90	NCTEQKKI	0
	99–112	DGSKNEWGWSKSKL	0
	142–150	QLAGKKSDQ	0
	260–290	YNSENKTYDLDFKTPNNDKSLVKTGAQLVDL	42

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
