# Peer review of "Targeting Plasmodium Life Cycle with Novel Parasite Ligands as Vaccine Antigens"

_vaccines, 2024, doi:10.3390/vaccines12050484_

Round 1

Reviewer 1 Report

Comments and Suggestions for Authors

Line 120, reference 27 is not adequate.

Line 181. "As human PLSCR1 ectodomain interacts with SARS-CoV-2 par-
ticles, it is expected that the PfPLSCR ectodomain will act as a ligand for hepatocyte interaction".  The rationale and conclusion are not clear

Line 183- "THEREFORE!!!!! none of the three predicted Plasmodium-specific B-cell epitopes in the N-terminal region (Table 2) is expected to generate protective antibodies. It is expected that the ligand epitope in the ectodomain may contribute for protective immunity when immunized". The rationale and conclusion are obscure.

Author Response

Line 120, reference 27 is not adequate.

Thanks for the correction. We changed the reference accordingly.

Line 181. "As human PLSCR1 ectodomain interacts with SARS-CoV-2 par-
ticles, it is expected that the PfPLSCR ectodomain will act as a ligand for hepatocyte interaction".  The rationale and conclusion are not clear

Thanks for the comment. We revised it.

Line 183- "THEREFORE!!!!! none of the three predicted Plasmodium-specific B-cell epitopes in the N-terminal region (Table 2) is expected to generate protective antibodies. It is expected that the ligand epitope in the ectodomain may contribute for protective immunity when immunized". The rationale and conclusion are obscure.

Thanks for the comment. We revised it.

Reviewer 2 Report

Comments and Suggestions for Authors

This review delineates antigens for use in malaria vaccines, predominantly based on the outstanding achievements of the authors' group. While progress in malaria vaccine research is not always substantial, the development of novel antigens such as those described herein holds significant importance. Moreover, the discovery of new interactions between the malaria parasite and its host is of great significance to biology. This review is considered highly beneficial not only as an introductory resource for researchers in malaria vaccine research but also for organizing existing knowledge in the field.

Reviewer recommends the publication of this review in Vaccines.

Minor comments are provided below:

1, Since the discussion mainly focuses on novel antigens, consider incorporating terms like "novel" into the title.

2, In the Abstract, it is stated (page 1, line 18) that current vaccines targeting the hepatic and mosquito stages do not focus on the ligand-receptor interactions between the host and parasite, which may not be entirely accurate. PfCSP and Pfs25, for instance, demonstrate such interactions.

3, On page 4, line 186, is "PLS" referring to PLSCR?

4, The interaction between PLSCR and CPSI is not clearly elucidated. It would be beneficial to provide further explanation regarding their relationship to infection. Is CPSI an enzyme of the urea cycle? Where does it localize, mitochondria perhaps? Not on cellular membrane?

Please consider these suggestions to refine and enhance the manuscript.

Author Response

1, Since the discussion mainly focuses on novel antigens, consider incorporating terms like "novel" into the title.

Thanks for the comment. Together with the editor’s suggestion we revised the title.

2, In the Abstract, it is stated (page 1, line 18) that current vaccines targeting the hepatic and mosquito stages do not focus on the ligand-receptor interactions between the host and parasite, which may not be entirely accurate. PfCSP and Pfs25, for instance, demonstrate such interactions.

PfCSP is important for sporozoite homing to the liver, however not a ligand. CSP interacts with negatively charged glycosaminoglycans (GAGs) that covers liver sinusoidal lining.  However, sporozoites can infect host cells without GAGs (Frevert et al., 1996. Mol. Biochem. Parasitol. 76:257–266).

Pfs25 is an important parasite surface protein for mosquito midgut invasion, however not a ligand. A parasite line missing Pfs25 still interacts with the mosquito midgut epithelium, however impaired in cell penetration (Baton et al., 2005., Malar. J., 4: 15).

3, On page 4, line 186, is "PLS" referring to PLSCR?

Thanks for the correction. We fixed it.

4, The interaction between PLSCR and CPSI is not clearly elucidated. It would be beneficial to provide further explanation regarding their relationship to infection. Is CPSI an enzyme of the urea cycle? Where does it localize, mitochondria perhaps? Not on cellular membrane?

Cha et al (2021. Nat Commun) identified CPS1 as a putative hepatocyte receptor since it interacts with sporozoite ligand PbPLSCR. CPS1 is a urea cycle enzyme which is abundant in hepatocyte mitochondria, however is secreted and exposed on the surface of mouse hepatocytes (Park et al. 2019. PNAS). We didn’t discuss on the function of the CPS1 since we think it’s out of focus.

Reviewer 3 Report

Comments and Suggestions for Authors

Comments vaccines 2961987

The MS “Targeting Plasmodium population bottlenecks with parasite 2 ligands as vaccine antigens” has been reviewed. Needless to say that malaria is a devastation illness for which vaccination can be the solution. It gives a very good overview of the subject and is well written. Therefore, I only have a few minor comments.

1)      Throughout the MS, it is to me not always clear of which species of Plasmodium they are talking.

2)      Line 124-126. Add a 3rd requirement: It can not be too similar to the host homolog.

3)      Line 208. For my own curiosity: Male gametocytes were exposed to phage-peptide display, but has that also been done with female gametocytes? Or would you in that case end up with the same ligand-receptor pair?

4)      Line 252. The abbreviation TBV was not used before. Please explain at first use.

Line 314-316: Just a remark: Isn’t it expected that, from the viewpoint of the parasite, the “bottleneck-stages” needs specially protection? Those stages are the weakest link in the chain. May be some tricks of the parasite can be expected when these ligands are used as a vaccine

Author Response

The MS “Targeting Plasmodium population bottlenecks with parasite 2 ligands as vaccine antigens” has been reviewed. Needless to say that malaria is a devastation illness for which vaccination can be the solution. It gives a very good overview of the subject and is well written. Therefore, I only have a few minor comments.

Thanks for the positive comments, however conflict with evaluations (1 star out of five in all criteria). Would you please consider modifying your star evaluations in the Review Report Form?

1)      Throughout the MS, it is to me not always clear of which species of Plasmodium they are talking.

We clarified it as P. falciparum in the end of Introduction (line 75).

2)      Line 124-126. Add a 3rd requirement: It can not be too similar to the host homolog.

Thanks for this comment. We added it.

3)      Line 208. For my own curiosity: Male gametocytes were exposed to phage-peptide display, but has that also been done with female gametocytes? Or would you in that case end up with the same ligand-receptor pair?

Phage display library was screened against purified male gametes, which identified MG1 peptide (Cha et al. 2024. mBio). Vega-Rodriquez et al (2015. Cell. Microbiol) screened the same library against purified female gametes, which identified FG1 peptide. Each screening identified peptides that bind to the most abundant surface molecule on each gamete, however not yet investigated whether they are in the same pathway.

4)      Line 252. The abbreviation TBV was not used before. Please explain at first use.

Thanks for the comment. We added a description in the Introduction. Malaria vaccines that target mosquito stage parasite development are called transmission-blocking vaccines (TBV).

Line 314-316: Just a remark: Isn’t it expected that, from the viewpoint of the parasite, the “bottleneck-stages” needs specially protection? Those stages are the weakest link in the chain. May be some tricks of the parasite can be expected when these ligands are used as a vaccine.

Thanks for the interesting idea!

Reviewer 4 Report

Comments and Suggestions for Authors

Khan et al reviewed important topics related to ligand from parasite that plays a detrimental role during Plamsodium transmission and can be potentially used as vaccine target. The authors focused on the ligands that were determined by peptide display library, either from yeast or phage. There are comprehensive results to support the potential ligands, but this may also limit the broaden of this review. The authors should revise the title and introduction to state the focus of this review is only on peptide display library. Or the author should include more findings not limited by peptide methods.

The authors offered detail review for each candidate and mentioned the use of candidate in other pathogens. But the information about the other pathogens sometimes distracts the reader. It may be easy to go through with a few sentences related the candidate in other pathogens. (Line 108-123, 236-243, 290-300) 

Line 34-39 Are there other mechanisms related this process?

Line 46 It have been shown that CSP antibody also interfere the skin stage of parasite.  (PMID: 30459199)

Line 77 There are other evidence that Kupffer is not the only cell related to exist of sporozoite. (PMID: 23610126)

Line 85 Ref 17 did not support the sentence.

Line 183 Ref 14 and 32 did not support the sentence.

Author Response

Khan et al reviewed important topics related to ligand from parasite that plays a detrimental role during Plasmodium transmission and can be potentially used as vaccine target. The authors focused on the ligands that were determined by peptide display library, either from yeast or phage. There are comprehensive results to support the potential ligands, but this may also limit the broaden of this review. The authors should revise the title and introduction to state the focus of this review is only on peptide display library. Or the author should include more findings not limited by peptide methods.

We agree that all the Plasmodium ligands during the two population bottlenecks were identified using phage peptide-display library technique, however no other ligands have been identified so far. Together with the reviewer 1 and editor’s suggestion we revised the title. However, we didn’t add ‘peptide display library’ since it may give readers bias to limit the scope of the review. Blood stage parasite ligands were identified using other approaches than peptide methods, however they are not involved in two population bottlenecks.

The authors offered detail review for each candidate and mentioned the use of candidate in other pathogens. But the information about the other pathogens sometimes distracts the reader. It may be easy to go through with a few sentences related the candidate in other pathogens. (Line 108-123, 236-243, 290-300) 

We added other cases besides malaria as evidences of successful application of moonlighting proteins against the concerns raised by the hypothesis in the ref 66. We agree these can distract the focus, however we also expect these cases will support activation of related research.

Line 34-39 Are there other mechanisms related this process?

This mechanism has been confirmed with diverse aspects and no major controversy has been reported yet.

Line 46 It have been shown that CSP antibody also interfere the skin stage of parasite.  (PMID: 30459199)

Thanks for the comment. We slightly change it.

Line 77 There are other evidence that Kupffer is not the only cell related to exist of sporozoite. (PMID: 23610126)

We already cited this in Line 40 (ref 5), however no related ligand and receptor for endothelial cell pathway has been identified.

Line 85 Ref 17 did not support the sentence.

Thanks for the comment. We fixed it.

Line 183 Ref 14 and 32 did not support the sentence.

Thanks for the comment. We deleted since they were already mentioned in advance.

Round 2

Reviewer 4 Report

Comments and Suggestions for Authors

The authors addressed most of my question.

To make the readers clear about the scope of this review, to clarify the focus about peptide-display library is informative in either title or abstract.  

Author Response

Thanks for the comment. We modified the end of the abstract (green highlight).